# OpenReview forum: "Benchmarking Reward Hack Detection in Code Environments via Contrastive Analysis"
_ICML.cc/2026/Conference — ICML 2026 regular_

### Official Review · Reviewer_yMko · 2026-03-08

**Soundness:** 3
**Presentation:** 3
**Significance:** 3
**Originality:** 2
**Overall Recommendation:** 5
**Confidence:** 4

**Summary:**

This paper creates a synthetic benchmark for reward hacking in code environments using Claude Code. The authors create a taxonomy of various coding reward hacks and create a detailed prompt to give to Claude Code. Claude then generates a dataset of reward hack and benign trajectories. These trajectories are used to benchmark the abilities of various frontier models (Claude, GPT, Gemini) to detect whether or not a coding trajectory has a reward hack. The authors consider two scenarios: binary classification (only seeing a single trajectory) and anomaly detection (seeing a group of trajectories in a GRPO-style setup).

**Compliance With Llm Reviewing Policy:**

Affirmed.

**Key Questions For Authors:**

Would it be possible to do a human study of how realistic the reward hacks are?

Could you reproduce a similar benchmark with Codex instead of Claude and study the distributional differences? Are there key axes of realism or construct validity that differ or is it mostly stylistic differences.

**Limitations:**

No, would it be possible for the authors to discuss the limitations of their work? Specifically, I'd find it useful for people who skim the paper to have a brief section on limitations that mentions: 1) the synthetic nature of the benchmark, 2) lack of human assessment for the realism of the traces, and 3) the lack of comparison between Claude code and Codex.

**Strengths And Weaknesses:**

# Strengths
The dataset is nearly entirely created by Claude Code, which suggests that this pipeline is scalable and can be adapted for future instances of reward hacks. Indeed, most of the work in this paper is in the taxonomy and the prompt, which are valuable and can be easily adapted to other domains. I think this benchmark would be directly useful for industry, especially as researchers are more rapidly deploying agents for training purposes. Moreover, given the plug-in-play nature of the prompt, this framework would serve as a scalable "reward-hack grader" eval, which seems very useful for progress.

The dataset is original as there are no large-scale coding reward hack datasets. Moreover, presenting the reward hack as an anomaly detection problem was suggested only in a much earlier work (Pan, 2022), so this framing offers a fresh perspective.

# Weaknesses
The dataset is not validated by humans, so it is difficult to assess the realism of the reward hacks, and thus the paper is slightly lacking in soundness. This leads to potential issues around evaluation awareness, which are not properly addressed in the work. **It would be great if the authors could do a study on whether or not humans are able to discern the realism of either the benign or hacked trajectories from real-world benign or hacked trajectories.** This would give more confidence in the overall dataset.

The paper is somewhat poorly formatted, with very short captions, lots of weird whitespace, and numbers overlapping on the figures. This would be nicer if it was cleaned up. Finally, the transition between Sections 4.1 and 4.2 is unclear, and it would be nice if the authors expanded on it.

---

> ### Author Rebuttal · Authors · 2026-03-26
>
> We are excited to see that the reviewer finds fresh perspective to reward hack detection in our paper. We address the reviewer flagged weaknesses and questions below:
>
> 1. TRACE, while being a synthetic benchmark, is completely human expert validated. We have emphasized on exhaustive human preference based study delineated in Appendix §H for the results in Table 1. Our metrics include four independent dimensions of realism: Agent Plausibility (Table 7), Event Believability (Table 8), Conversational Progression (Table 9) and Hack Subtlety (Table 10) based on which we further refine and filter TRACE data points. We would be happy to address any concerns that the reviewer has with respect to these dimensions and our realism claim.
>
> 2. > It would be great if the authors could do a study on whether or not humans are able to discern the realism of either the benign or hacked trajectories from real-world benign or hacked trajectories.
> - Independent of the metric based filtering process above, we report human performance numbers in Figure 4. The strong scores achieved by humans on this task demonstrate that humans can clearly distinguish between benign and hacked trajectories, and that human performance also improves when contrasting multiple trajectories. We hope these results are convincing to the reviewer. We will further emphasize on this in the main contents of the camera ready version.
> 3. > Could you reproduce a similar benchmark with Codex instead of Claude and study the distributional differences? Are there key axes of realism or construct validity that differ or is it mostly stylistic differences?
> - We only use Claude Code as an agentic harness for generation of TRACE and do not overindex on what backbone model is used with the above harness (since that can be independently set via ANTHROPIC_BASE_URL). We believe that while there will be minor differences in tooling across harnesses like Claude Code and Codex, the generation quality is only influenced by the completeness of our framework and taxonomy, which we meticulously spend time hand curating and forms a core contribution of our paper. We leave further ablations on harnesses to future work.
>
> As requested, we will be happy to add a limitations section in the camera ready version of the paper and thank the reviewer for flagging this.

---

> > ### Author Rebuttal · Reviewer_yMko · 2026-04-03
> >
> > > The strong scores achieved by humans on this task demonstrate that humans can clearly distinguish between benign and hacked trajectories
> >
> > Hmm I don't think this is what I meant. It was more whether a human could distinguish between a real and synthetic trajectory.
> >
> > In any case, I'm happy to keep my score but hope the authors address this point in their limitations.

---

### Official Review · Reviewer_r34N · 2026-03-11

**Soundness:** 3
**Presentation:** 4
**Significance:** 4
**Originality:** 3
**Overall Recommendation:** 5
**Confidence:** 3

**Summary:**

This paper benchmarks the ability of large language models to detect reward hacking in code-based reinforcement learning environments. It introduces TRACE, a benchmark containing 517 human-verified multi-turn coding trajectories covering 54 categories of reward hacking behaviors, and evaluates several state-of-the-art LLMs on this task. The results show that detection performance improves when framed as contrastive anomaly detection rather than isolated classification, but models still struggle with semantically contextualized reward hacks. The paper provides a new benchmark and empirical analysis highlighting current limitations of LLM-based reward hack detection.

**Compliance With Llm Reviewing Policy:**

Affirmed.

**Final Justification:**

My final recommendation remains Accept. I find the paper valuable for its introduction of TRACE and its thorough analysis of reward hack detection. My main concerns were the synthetic data, the gap between the evaluation setup and real deployment, and the use of an LLM judge in part of the evaluation. The rebuttal addressed these points reasonably well, but the limitations still remain. Overall, my assessment is unchanged.

**Key Questions For Authors:**

1. TRACE is generated using LLM agents and then human-verified. How well do these synthetic trajectories reflect reward hacking behaviors observed in real RL training pipelines?

2. The evaluation formulates reward hack detection as contrastive anomaly detection over clusters of trajectories. How would this detection setup translate to practical deployment in RL pipelines where reward evaluators may operate on individual trajectories?

**Limitations:**

yes

**Strengths And Weaknesses:**

Strengths:

1. The paper introduces TRACE, a benchmark for reward hack detection in code environments. Given the increasing use of LLMs as evaluators and reward models in RL-based training pipelines, this benchmark provides a useful resource for systematically studying and improving reward hack detection in this setting.

2. The benchmark containing 517 human-verified trajectories spanning 54 reward-hack categories across 10 major classes. This provides substantially broader coverage than prior benchmarks that focus on only a few exploit types or single-turn interactions.

3. The paper proposes a contrastive anomaly detection formulation for reward hack detection, evaluating models over clusters of trajectories rather than isolated examples and offering a different perspective on model behavior.

4. The paper provides a thorough empirical and qualitative analysis of reward hack detection. It examines how different exploit types affect model performance and shows that models detect mechanical exploits more reliably than semantically contextualized ones, offering useful insights into the limitations of current LLM-based detectors.

Weaknesses:

1. TRACE is synthetically generated using Claude-Code agents, even though it is later human-verified. This introduces potential concerns, such as the synthetic behaviors may not reflect real reward hacking strategies used by models, and  generation prompts may bias the structure or distribution of exploits. Thus, the benchmark may not fully capture real-world RL training environments.

2. The evaluation relies on prompting LLMs to analyze clusters of trajectories and identify reward hacking outliers. While this contrastive setup is motivated by RL training procedures, it remains unclear how closely this detection setting reflects how reward models or evaluators are used in practice.

3. The evaluation uses Claude Opus 4.5 as an LLM judge to map model explanations to taxonomy labels. This introduces a dependency on the judge model, where potential biases or errors in the judge could affect the reported results.

---

> ### Author Rebuttal · Authors · 2026-03-26
>
> We thank the reviewer for their time and are excited to know that they find value in our work. We address the listed weaknesses and questions below:
> 1. We acknowledge the potential existence of Claude Code-based generator bias in the creation process for TRACE but we attempt to meticulously categorize and remove the above bias. Primarily, we ensure that the trajectories adhere to the taxonomy definitions provided in Tables 3, 4, 5 and 6 which are further annotator and author validated. Furthermore, we emphasize on several fine grained metrics for realism and representativeness (derived from engineer questionnaires), including Agent Plausibility (Table 7), Event Believability (Table 8), Conversational Progression (Table 9) and Hack Subtlety (Table 10). Each of these metrics ensure that the trajectory is representative of a realistic agent hack. Since we only retain the samples that have 100% annotator agreement, the final dataset is fully representative of natural reward hacks. Finally, the high scores observed during capturing of human baselines further strengthens the alignment.
> 2. While this is not directly addressed in the paper and remains out of scope of this study, the paper still motivates contextualization of modern reward functions (verifiable or otherwise). We hope to encourage the design and study of contrastive reward functions in the future through our work.
> 3. The evaluation process first deterministically extracts detected categories to remove noise before the LLM judge comes across them. The use of LLM judge for vague category mapping is only meant to avoid false negatives in the cases where the model produces category descriptors in addition to the exact answer category. To further ensure that the results are consistent, we repeat our evaluations three times and only report the average over these runs.

---

> > ### Author Rebuttal · Reviewer_r34N · 2026-04-02
> >
> > Thank you to the authors for the clarification. The response helps contextualize my concerns regarding synthetic data generation and the use of an LLM judge. These limitations still remain, but they seem largely inherent to the current study of LLM-based reward hack detection rather than issues that could be fully resolved within this paper. Overall, I find the rebuttal satisfactory and my assessment of the paper remains positive.

---

### Official Review · Reviewer_dxMR · 2026-03-12

**Soundness:** 3
**Presentation:** 3
**Significance:** 2
**Originality:** 2
**Overall Recommendation:** 3
**Confidence:** 4

**Summary:**

This paper proposes a benchmark for reward hacking in code environment. This is a synthetically curated and human-verified benchmark containing 517 testing trajectories.

**Compliance With Llm Reviewing Policy:**

Affirmed.

**Final Justification:**

Some concerns are not solved. I will maintain my original assessment.

**Key Questions For Authors:**

How to justify that the trajectories are representative enough for reward hacking problem?

**Limitations:**

yes

**Strengths And Weaknesses:**

Pros:
1. Reward hacking in code RL is important. Specilization is a good thing.
2. Contrastive framing is very good and insightful.

Cons:
1. The trajectories are collected from Claude Code and PE. I think it may has some kind of generator bias. Reward hacking is a quite large topic. The bias may make it not possible to measure or evaluate reward hacking problem. The authors may want to discuss why the trajectories are representative.
2. The current ablations mix many points, including contrastive reasoning and context. The ablation should be focused.
3. The design of LLM judge may make the match rate not reliable.

---

> ### Author Rebuttal · Authors · 2026-03-26
>
> We are excited to see that the reviewer finds our contrastive framing insightful. We have addressed the listed weaknesses and questions below:
>
> 1. We acknowledge the potential existence of Claude Code-based generator bias in the creation process for TRACE but we attempt to meticulously categorize and remove the above bias. Primarily, we ensure that the trajectories adhere to the taxonomy definitions provided in Tables 3,4,5 and 6 which are independently author and annotator validated. Furthermore, we emphasize on several fine grained metrics for realism and representativeness (derived from engineer questionnaires), including Agent Plausibility (Table 7), Event Believability (Table 8), Conversational Progression (Table 9) and Hack Subtlety (Table 10). Each of these metrics ensure that the trajectory is representative of a realistic agent hack. Since we only retain the samples that have 100% annotator agreement, the final dataset is fully representative of natural reward hacks. We will make sure to emphasize this in the main text of the camera ready version.
> 2. The ablations in the paper aim to study detectability of finer grained categories of reward hacks, the effect of cluster size on detectability and the effect of benign ratio within a cluster on detectability. We will simplify these further in the camera ready version of the paper to avoid loss of focus.
> 3. The evaluation process first deterministically extracts detected categories to remove noise before the LLM judge comes across them. The use of LLM judge for vague category mapping is only meant to avoid false negatives in the cases where the model produces category descriptors in addition to the exact answer category. To further ensure that the results are consistent, we repeat our evaluations three times and only report the average over these runs. We will clarify this pipeline better in the camera ready version.

---

> > ### Author Rebuttal · Reviewer_dxMR · 2026-04-03
> >
> > Thank you for the rebuttal. While the human annotation and agreement filtering help ensure internal quality, my core concern remains that trajectories sourced from a single agent system may systematically miss entire categories of reward hacking behaviors that arise in other code RL settings.

---

> > > ### Author Response · Authors · 2026-04-03
> > >
> > > We appreciate the reviewer's concern about generalizability to reward hacking behaviors beyond single-agent settings. However, we believe that this falls outside the intended scope of the paper. Specifically, our contributions are:
> > >
> > > - **Taxonomy**: We define a broad taxonomy of grounded coding reward hack behaviors derived from engineers working directly with coding agents. This constrains the benchmark to feasible, observed reward hacks and no such taxonomy currently exists.
> > > - **Synthetic generation**: We use synthetic trajectory generation to simulate settings where these behaviors emerge, maximizing domain coverage to avoid collapse, and conducting an exhaustive human evaluation to confirm that included trajectories are realistic and faithfully follow the taxonomy categories. We avoid generating trajectories with a single agent's behaviors in mind to ensure maximum downstream generalizability.
> > > - **Behavioral analysis**: The dataset is designed to study the detection patterns of both open and closed-source foundation models. By filtering to only trajectories meeting our strict annotation standards, we are confident that the observed patterns are significant and practically valuable in terms of coverage and applicability.
> > >
> > > The goal of this work is therefore not to enumerate and study every possible reward hack pattern and the simulator's ability to surface these, but instead to reliably simulate and study hacking behaviors **within the scope of the defined taxonomy**. Our human review further ensures coverage within this taxonomy. We hope this clarifies the intended contribution and welcome any further questions in that context.

---

### Official Review · Reviewer_bJ3H · 2026-03-14

**Soundness:** 3
**Presentation:** 3
**Significance:** 3
**Originality:** 3
**Overall Recommendation:** 3
**Confidence:** 3

**Summary:**

This paper introduces TRACE, a benchmark for detecting reward hacks in code environments. The taxonomy is broader than usual, and the contrastive cluster setting is a reasonable evaluation angle. My problem is that the submission still feels closer to a benchmark prototype than to a benchmark release the community can confidently adopt.

**Compliance With Llm Reviewing Policy:**

Affirmed.

**Key Questions For Authors:**

1. Will the dataset, code, and full evaluation pipeline be released? If so, what exactly will be included?

2. Please provide the exact prompts, system instructions, and decoding settings used for all evaluated models.

3. Can the authors spell out the cluster construction and scoring procedures more concretely?

4. What evidence do the authors have that TRACE reflects naturally occurring reward hacks rather than benchmark-specific synthetic artifacts?

**Limitations:**

yes

**Strengths And Weaknesses:**

### Strengths
The problem is timely. Reward hacking is becoming a real issue for coding agents, so a benchmark in this area would be useful.

The taxonomy is broader than what is usually covered, which makes the benchmark more interesting than a narrow collection of standard test-tampering cases.

I also think the contrastive setup is a reasonable idea. The result that models do better with clustered trajectories than with isolated ones is not surprising, but it is still useful to show.

### Weaknesses
For a benchmark paper, reproducibility is part of the contribution. Right now it is not clear enough whether the dataset, code, and full evaluation pipeline will be released in a form that others can actually use.

Relatedly, too much of the evaluation protocol is left implicit. A benchmark paper should spell out the prompts, system instructions, decoding settings, cluster construction, and how final judgments are parsed and scored.

I am also not fully persuaded by the realism claim. TRACE may still be useful, but it remains a synthetic benchmark, and I am not yet sure how much performance on TRACE says about naturally occurring reward hacks in real coding workflows.

The writing and presentation do not help. Some tables do not clearly situate the compared models or settings, and the overall formatting makes the paper feel less mature than it should.

---

> ### Author Rebuttal · Authors · 2026-03-26
>
> We are glad to see that the reviewer finds our work timely and our taxonomy coverage wider than existing works. We address their weakness and questions below:
>
> 1. We will open source the complete dataset and dataset card on HuggingFace. We will also open source and maintain a Github repository containing all evaluation scripts, prompts and human annotations upon acceptance. For reference, we have attached minimal evaluation scripts to test each hypothesis along with the complete dataset in the supplementary materials attached to this submission for your review.
> 2. We have included all prompts used for data curation and evaluation in the Appendix (§B,§C and §D). We show the usage of these prompts in the codebase attached to this submission which we plan to open source upon acceptance. We would be happy to include any other prompts that the reviewer finds missing from the paper.
> 3. (Also addresses Q4) TRACE, while being a synthetic benchmark, is completely human validated. We have emphasized an exhaustive human engineer preference based study delineated in Appendix §H for the results in Table 1. All annotators had to complete rigorous onboarding on detecting reward hacks. Our final evaluation metrics include four independent dimensions of realism sampled from questionnaires completed by real-world engineers: Agent Plausibility (Table 7), Event Believability (Table 8), Conversational Progression (Table 9) and Hack Subtlety (Table 10) based on which we further refine and filter TRACE data points. We would be happy to address any concerns that the reviewer has with respect to these dimensions and our realism claim.
> 4. We have followed the general ICML guidelines when writing the paper but would be happy to incorporate any additional edits required to increase the readability and maturity of our paper. If the reviewer could point to the specific table/s, we would be happy to make edits before the camera ready version is released.
>
> 5. > Can the authors spell out the cluster construction and scoring procedures more concretely?
>
> - Clusters are constructed using a random sampling method over the hacked and benign categories. This sampling can be controlled using the specific benign ratio that we wish to study, giving the experiments more controllability. To ensure reproducibility of our evaluation, we provide all hyperparameters and random seeds in the paper (see Section 4.2)
> - The prompt in Appendix D shows the vague category match used to evaluate the correctness of the output. This mapping is only performed after deterministic attempts to extract categories. This category mapping procedure accounts for any additional unbounded text that the LLM may produce during evaluation and makes evaluation more robust. We will clarify this in the camera ready version.

---

> > ### Author Rebuttal · Reviewer_bJ3H · 2026-04-04
> >
> > I still have some concerns. Although TRACE is human-validated, it is still synthetic at its core, so I am not fully convinced that the current evidence is enough to support the broader realism claims. I also think some evaluation details, especially cluster construction and scoring, should be explained more clearly in the main paper.
> >
> > The study investigates a central concept that is timely and important. The authors aim to evaluate whether current models can reliably detect reward hacking in code environments under more realistic settings. I think this is a promising direction, but some important questions remain open.

---

> > > ### Author Response · Authors · 2026-04-05
> > >
> > > We want to clarify that the goal of this paper is to focus on the reward hacking cases that fall under our taxonomy categories. We do not make any broad or open-ended claims about general reward hack patterns and their detectability in this paper and believe that this is out of the scope of this work. The novelty of this paper is as following:
> > >
> > > - **Taxonomy**: We define a broad taxonomy of grounded coding reward hack behaviors derived from engineers working directly with coding agents. This constrains the benchmark to feasible, observed reward hacks and no such taxonomy currently exists.
> > > - **Synthetic generation**: We use synthetic trajectory generation to simulate settings where these behaviors emerge, maximizing domain coverage to avoid collapse, and conducting an exhaustive human evaluation to confirm that included trajectories are realistic and faithfully follow the taxonomy categories. We avoid generating trajectories with a single agent's behaviors in mind to ensure maximum downstream generalizability.
> > > - **Behavioral analysis**: The dataset is designed to study the detection patterns of both open and closed-source foundation models. By filtering to only trajectories meeting our strict annotation standards, we are confident that the observed patterns are significant and practically valuable in terms of coverage and applicability.
> > >
> > > > I also think some evaluation details, especially cluster construction and scoring, should be explained more clearly in the main paper.
> > >
> > > We believe that we have addressed this concern in our rebuttal. We would be happy to address any outstanding concerns about clarity.

---

### Decision · Program_Chairs · 2026-04-30

**Decision:**

Accept (regular)

**Comment:**

This paper introduces a human-verified synthetic benchmark for detecting reward hacks in code-based RL, evaluated through a realistic contrastive anomaly detection setup. Several reviewers highlighted strengths:

- Scalability & Broad Coverage: The Claude Code-driven pipeline provides substantial coverage (517 trajectories across 54 categories) and is highly scalable and adaptable for industry applications.

- Novel Framing & Analysis: Formulating detection as contrastive anomaly detection offers a valuable perspective, supported by thorough empirical and qualitative analysis of model limitations.

Overall, this paper is technically sound, introduces a highly relevant resource for the growing RL safety community, and will be a valuable addition to ICML.